# Mathematical modelling of mechanotransduction via RhoA signalling pathways

**Sofie Verhees** [iD][1][☯][*], **Chandrasekhar Venkataraman**[2][☯], **Mariya Ptashnyk**[1][☯]

**1** Department of Mathematics, Heriot-Watt University, The Maxwell Institute for Mathematical Sciences, Edinburgh, United Kingdom, **2** Department of Mathematics, University of Sussex, Brighton, United Kingdom

☯ These authors contributed equally to this work.

* sbv2000@hw.ac.uk

**Data availability statement:** The code and data used to produce the results in this manuscript are available at https://github.com/sofieverhees/ model-mechanotransduction.

## Abstract

We derive and simulate a mathematical model for mechanotransduction related to the Rho GTPase signalling pathway. The model addresses the bidirectional coupling between signalling processes and cell mechanics. A numerical method based on bulk-surface finite elements is proposed for the approximation of the coupled system of non-linear reaction-diffusion equations, defined inside the cell and on the cell membrane, and the equations of elasticity. Our simulation results illustrate novel emergent features such as the strong dependence of the dynamics on cell shape, a threshold-like response to changes in substrate stiffness, and the fact that coupling mechanics and signalling can lead to the robustness of cell deformation to larger changes in substrate stiffness, ensuring mechanical homeostasis in agreement with experiments.

## Author summary

Mechanotransduction, a process by which cells convert mechanical stimuli into chemical signals, plays a crucial role in cell functions. To better understand this phenomenon, we need to analyse how signalling processes and cell mechanics work together. For this purpose we derive and simulate a mathematical model of mechanotransduction related to the Rho GTPase signalling pathways, central to almost all fundamental cellular processes including cell polarity, movement, division, and cytoskeleton reorganization. The model introduces a two-way coupling between the signalling processes and cell mechanics. We use a numerical method based on bulk-surface finite elements to solve model equations numerically. Our simulation results illustrate novel emergent features such as a strong dependence of the dynamics on cell shape, a threshold-like response to changes in substrate stiffness, and the fact that the two-way coupling between mechanics and signalling can lead to the robustness of cell deformation to larger changes in substrate stiffness,

**Funding:** SV was supported by the EPSRC Centre for Doctoral Training in Mathematical Modelling, Analysis and Computation (MAC-MIGS) funded by the UK Engineering and Physical Sciences Research Council (grant EP/S023291/1), Heriot-Watt University and the University of Edinburgh. CV acknowledges support from the Dr Perry James (Jim) Browne Research Centre on Mathematics and its Applications (University of Sussex). The funders had no role in study design, data collection and analysis, decision to publish, or preparation of the manuscript.

**Competing interests:** The authors have declared that no competing interests exist.

ensuring mechanical homeostasis in agreement with experiments. These interesting insights help us unravel the underlying mechanisms in mechanotransduction.

## Introduction

Intercellular signalling processes constitute the mechanisms through which cells communicate with and respond to their environment. Hence, signalling pathways are important in all physiological activities of the cell, such as cell division, cell movement, the immune response, and tissue development [1]. Aberrant cell signalling can often result in the development of diseases [2]. It is therefore important to understand signalling phenomena. Recent studies have found that alongside biochemical reactions, mechanics plays an important role in many signalling pathways [3,4]. This phenomena is referred to as mechanotransduction which, broadly speaking, is any process by which cells convert mechanical stimuli into chemical signals [5,6].

A large number of recent works study the role of Rho GTPases, primarily RhoA, in mechanotransduction in relation to different mechanical cues: extracellular matrix (ECM) stiffness and viscoelasticity, tensile stress (stretching), compressive stress (compression), and shear stress (fluid flow shear), see e.g. [7,8] for a review. Moreover, the coupling between biochemistry and mechanics is bidirectional, i.e., chemical signals can also affect the mechanical properties of the cell, such as molecules like focal adhesion kinases (FAKs) that influence F-actin dynamics and therefore the stiffness of the cell [6,9–11].

The formidable complexity of the phenomena involved in mechanotransduction means that much about how the mechanics and the chemical processes of the cell communicate is not yet understood and mathematical modelling is crucial in this regard. Whilst the mathematical modelling of biochemical cell signalling processes is fairly well developed, e.g., [12–14], the study of mechanotransduction is comparatively more recent, see [15] for a review. Typically the modelling involves solving coupled systems of partial differential equations (PDEs) with reaction-diffusion equations modelling the biochemistry coupled to equations based on (visco)elastic constitutive laws for the mechanics. The progress of such efforts has been rapid, ranging from early models employing simplifications such as one-dimensional geometries [16,17] to full three-dimensional simulations [18] using advanced computational techniques. Alongside continuum models, a number of recent works have employed discrete approaches such as spring-based models [19], or models that employ a Potts formalism [20–22]. Despite this rapid progress, the existing models typically make major simplifying assumptions such as assuming a constant stiffness of the ECM [18,23,24], as well as neglecting the two-way coupling in which signalling pathways affect the mechanics alongside mechanical cues inducing signalling processes.

In the present work, we seek to develop, analyse and simulate a model for mechanotransduction through the Rho GTPase signalling pathway which allows for a two-way coupling between the mechanics and the biochemistry. The dynamics of the signalling molecules FAK and RhoA are modelled using reaction-diffusion equations, where the ECM stiffness and elastic stresses of the cell activate FAK. Under simplifying assumptions, i.e., assuming no dependence on the cell elastic stresses, the biochemical component of the model is derived as a reduction of the model proposed in [18]. For the cell's mechanical properties, we assume an elastic constitutive relationship [25] and allow the material properties to depend on the concentrations of the signalling molecules. We propose a numerical method based on bulk and surface finite elements [26] for the approximation of the model equations.

The results presented here show that our model can reproduce the qualitative results of [18], i.e., the mass of activated FAK and RhoA depend on ECM stiffness, with the dependence captured by a Hill function. On the inclusion of the two-way coupling between signalling processes and cell mechanics, we observe novel dynamics, such as the conservation of cell deformation under different values of the ECM stiffness, which underlines the importance of including these more complex models of the mechanics. The role of mechanotransduction in homeostasis in biological processes has been discussed in a number of biological works, e.g., [4,5,9,10,27] and our work presents a concrete example of how modelling can help elucidate potential mechanisms that underlay the mechanical homeostasis. Homeostasis of cell deformation, as observed in simulations of our model, has been observed experimentally [28]. Our focus is on elastic constitutive assumptions for the mechanics of the cell to enhance clarity of exposition and to avoid unnecessary technical complexities. This can be extended to allow for other constitutive laws such as viscoelasticity of the cell and/or of the ECM as has been done elsewhere in the literature in simpler settings in 1D [16,29]. This work thus serves as a starting point in modelling and analysis of the two-way coupling between mechanics and chemistry.

The paper is organised as follows. We first derive the reduced model for the Rho GTPase signalling pathway, based on the model proposed in [18]. Next, the mathematical model for the mechanotransduction related to the Rho GTPase signalling pathway is derived. Simulations of the model are presented in the section after. We conclude the paper with a discussion of the results. Details on the numerical method applied to simulate the model equations are given in S1 Appendix, Sect A.3.

## Methods

### A mathematical model for the Rho GTPase signalling pathway

One of the main signaling pathways involved in mechanostranduction is the Rho GTPase pathway, responsible for many important cellular processes, e.g. motility, cell adhesion, polarisation, differentiation, remodelling of the exoskeleton, and the ECM [8]. The RhoA signalling pathway is activated through the activation of FAK in response to tension on integrins, which depends on ECM stiffness [1,11], see Fig 1 for an overview.

Our model for mechanotransduction related to the RhoA-mediated intercellular signalling pathway is based on models developed in [18] and [23,24]. To incorporate the interactions between mechanics and signalling processes, we extend the model proposed in [18] by considering elastic deformations of the cell. Activated FAK is downstream in the RhoA GTPase signalling pathway and hence the activation of RhoA is a function of activated FAK. The activation of RhoA results in ECM remodelling and deposition of new fibres, increasing ECM stiffness and hence activation of FAK [30]. FAK is expressed in the cytoplasm of the cell and is activated on the cell membrane. To simplify the model and focus only on the most significant aspects from the perspective of qualitative behaviour, we reduce the model for the RhoA signalling pathway of [18] that includes the dynamics of FAK, RhoA, ROCK, Myo, LIMK, mDia, Cofilin, F-actin and YAP/TAZ by considering only the dynamics of FAK and activated RhoA. Such a reduction is possible since other molecules considered in the full model of [18] do not influence the dynamics of FAK and RhoA. Our rationale behind considering a simplified model is to more clearly elucidate the emergent features that arise when mechanics is coupled with signalling. It is not challenging to incorporate other biochemical species or different reaction kinetics within the framework we propose.

We let $Y \subset \mathbb{R}^3$ denote the cytoplasm and $\Gamma = \partial Y$ the cell membrane. We denote by $\phi_d$ and $\phi_a$ the concentrations of inactive and active FAK, and by $\rho_a$ the concentration of active RhoA.

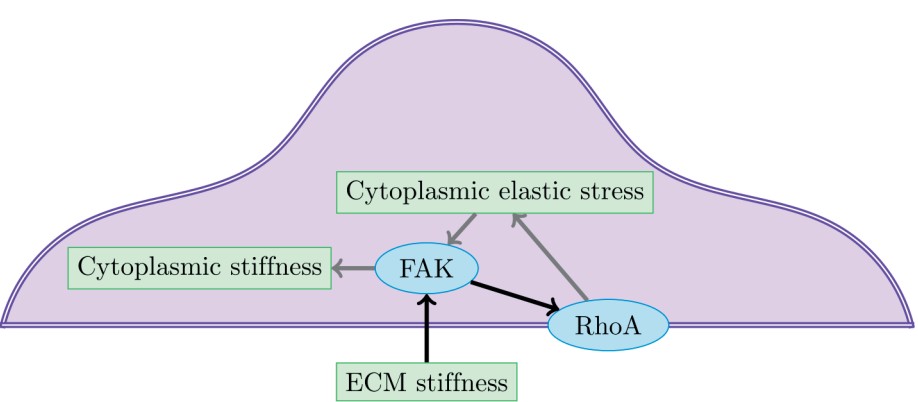

**Fig 1. A sketch of the model interactions.** The diagram shows a simplified overview of the interactions in the models presented in this paper, where the blue ellipses represent the signalling molecules and the green rectangles the mechanical signals. The black arrows represent the reactions of the biochemical model in Eq (1) based on [18], i.e. ECM stiffness activates FAK on the cell membrane which activates RhoA. The grey arrows represent the coupling between chemistry and mechanics introduced in this work and described by Eqs (3)-(6), i.e. activated RhoA affects the cytoplasmic elastic stress which activates FAK and the cytoplasmic stiffness is a function of activated FAK.

We recall that FAK, both active and inactive, is assumed to be cytoplasm resident and activated RhoA membrane resident. Our reduced model for the biochemistry consists of the following system of reaction-diffusion equations

$$
\begin{aligned}
\partial_t \phi_d - D_1 \Delta \phi_d &= k_1 \phi_a && \text{in } Y,\ t > 0, \\
\partial_t \phi_a - D_2 \Delta \phi_a &= -k_1 \phi_a && \text{in } Y,\ t > 0, \\
D_1 \nabla \phi_d \cdot \nu &= -n_r k_2 \phi_d - n_r k_3 \frac{E}{C+E} \phi_d && \text{on } \Gamma,\ t > 0, \\
D_2 \nabla \phi_a \cdot \nu &= n_r k_2 \phi_d + n_r k_3 \frac{E}{C+E} \phi_d && \text{on } \Gamma,\ t > 0, \\
\partial_t \rho_a - D_3 \Delta_\Gamma \rho_a &= -k_4 \rho_a + n_r k_5 \left( (\gamma \phi_a)^n + 1 \right) \left( \frac{M_\rho}{|Y|} - \frac{\rho_a}{n_r} \right) && \text{on } \Gamma,\ t > 0, \\
\phi_d(0,x) &= \phi_d^0(x), \quad \phi_a(0,x) = \phi_a^0(x) \quad \text{in } Y, \qquad \rho_a(0,x) = \rho_a^0(x) \quad \text{on } \Gamma,
\end{aligned}
$$

(1)

where $\Delta_\Gamma$ is the Laplace Beltrami operator modelling diffusion on the surface $\Gamma$, see e.g. [26], $n_r = |Y|/|\Gamma|$ is the ratio between the volume of the cytoplasm and the area of the cell membrane, $k_1, k_4$ are deactivation and $k_2, k_3, k_5$ are activation constants, $E$ is the substrate stiffness, $D_1, D_2, D_3$ are the diffusion constants, $C, n$ and $\gamma$ are positive constants, $\phi_d^0(x), \phi_a^0(x)$ and $\rho_a^0(x)$ are bounded nonnegative functions, and $\frac{M_\rho}{|Y|} - \frac{\rho_a}{n_r}$ is an approximation for deactivated RhoA ($\rho_d$) on the surface with $M_\rho = \int_Y \rho_d^0 \mathrm{d}x + \int_\Gamma \rho_a^0 \mathrm{d}s$ the total mass of RhoA, a quantity conserved in the full model of [18] and assumed to be conserved here. Simulations illustrating the agreement between results obtained using the reduced model (1) with those of [18] for the full model are presented in S1 Appendix, Sect A.1.

## Mathematical model for mechanotransduction

As a starting point for the mechanics, we consider small deformations and hence, assume a linear elastic constitutive law for the mechanics of the cell. Although viscoelastic or poroelastic behaviour of cells is proposed in many works [31,32], linear elasticity is often chosen

for modelling simplicity and it can yield results consistent with experimental observations [33–35]. Our focus is modelling a bidirectional coupling between cell stiffness and signalling processes, a minimal model assuming a linear elastic law for the cell mechanics is therefore sufficient for this work, as it avoids unnecessary complications that arise from the consideration of viscous stresses. To demonstrate that our results remain relevant under more complex assumptions, a linear viscoelastic model is presented and simulated in Sect A.7 in S1 Appendix. We note that for the corresponding simulation results, the qualitative behaviour in both cases, linear elastic and linear viscoelastic, is the same.

An important simplification that arises under the small deformations assumption inherent in this work is that the dynamics of the signalling molecules may be effectively considered on the reference configuration with no additional terms arising due to the deformation. Models where the assumption of small deformations is relaxed will be addressed in future studies.

The cell nucleus plays an important role in governing the mechanical properties of the cell [36,37], whilst we predominantly neglect this in the present work, in S1 Appendix, Sect A.5, we have included simulations of the model with a 'passive' nucleus that is considered to be more rigid than the cytoplasm.

It has been shown that the stiffness of the cell increases as F-actin increases [18]. F-actin is downstream from activated FAK and, as apparent from the results in [18], we can use activated FAK as a proxy for F-actin. Therefore, we assume that the Young's modulus $E_c$ of the cell is a function of the activated FAK concentration. Based on experimental observations [38] and numerical simulations [18], we propose

$$E_c = E_c(\phi_a) = k_7\big(1 + (k_8\phi_a)^p\big), \tag{2}$$

where $k_7$, $k_8$ and $p$ are non-negative constants. Then for elastic deformations of the cell, we have

$$-\nabla \cdot \sigma(u) = 0 \quad \text{in } Y, \tag{3}$$

with

$$\sigma(u) = \lambda(\phi_a)(\nabla \cdot u)I + 2\mu(\phi_a)\big(\nabla u + (\nabla u)^T\big)$$

and the Lame constants $\lambda$ and $\mu$ are given by

$$\lambda(\phi_a) = \frac{E_c(\phi_a)\nu_c}{(1 + \nu_c)(1 - 2\nu_c)}, \quad \mu(\phi_a) = \frac{E_c(\phi_a)}{2(1 + \nu_c)},$$

where $\nu_c$ is the Poisson ratio of the cell.

Activated RhoA regulates remodellling of stress fibres inside the cell and stabilisation of actin filaments [39–41,43]. This mechanism is modelled by the stress on the boundary being dependent on activated RhoA concentration

$$\sigma(u)\nu = k_6\mathbb{P}(\rho_a\nu) \quad \text{on } \Gamma, \tag{4}$$

where $k_6$ is a positive constant, $\mathbb{P}$ is a projection on the space orthogonal to the space of rigid deformations, i.e. rotations and translations. Alongside models where the cell is allowed to

deform freely, to model a typical experimental set-up where cells are placed on a rigid substrate, we consider

$$u \cdot \nu = 0, \qquad \Pi_\tau(\sigma(u)\nu) = 0 \qquad \text{on } \Gamma_0, \tag{5}$$

together with condition Eq (4) on $\Gamma \setminus \Gamma_0$, where $\Pi_\tau(w) = w - (w \cdot \nu)\nu$ denotes the tangential projection of vector $w$, and hence $\Pi_\tau(\sigma(u)\nu)$ is the shear stress and Eq (5) specifies a shear stress free condition. In this work, we choose $\Gamma_0 = \Gamma \cap \{x \in \mathbb{R}^3 | x_3 = 0\}$ to simulate a rigid substrate.

It has been shown that an increased contractility is associated with increased activated FAK, see e.g. [39]. Thus we assume that FAK is activated by the stress of the cell and as a proxy for the cytosolic stress we use the positive part of trace of the Cauchy stress tensor $\text{tr}(\sigma)_+$, where $\text{tr}(\sigma)$ is the first stress invariant and the positive part reflects the fact that extension rather than compression causes the activation of FAK. This modifies the system in Eq (1) to

$$\begin{aligned}
\partial_t \phi_d - D_1 \Delta \phi_d &= \quad k_1 \phi_a - C_1 \text{tr}(\sigma)_+ \phi_d & &\text{in } Y, \, t > 0, \\
\partial_t \phi_a - D_2 \Delta \phi_a &= -k_1 \phi_a + C_1 \text{tr}(\sigma)_+ \phi_d & &\text{in } Y, \, t > 0, \\
D_1 \nabla \phi_d \cdot \nu &= -n_r k_2 \phi_d - n_r k_3 \frac{E}{C+E} \phi_d & &\text{on } \Gamma, \, t > 0, \\
D_2 \nabla \phi_a \cdot \nu &= \quad n_r k_2 \phi_d + n_r k_3 \frac{E}{C+E} \phi_d & &\text{on } \Gamma, \, t > 0, \\
\partial_t \rho_a - D_3 \Delta_\Gamma \rho_a &= -k_4 \rho_a + n_r k_5 \left((\gamma \phi_a)^n + 1\right) \left(\frac{M_\rho}{|Y|} - \frac{\rho_a}{n_r}\right) & &\text{on } \Gamma, \, t > 0, \\
\phi_d(0,x) = \phi_d^0(x), \quad \phi_a(0,x) &= \phi_a^0(x) \quad \text{in } Y, \qquad \rho_a(0,x) = \rho_a^0(x) & &\text{on } \Gamma,
\end{aligned} \tag{6}$$

where $\nu_+ = \max\{\nu, 0\}$. We can prove existence, uniqueness and boundedness of solutions to the system in Eqs (3)–(6) which we intend to report on elsewhere.

## Initial and boundary conditions and parametrization

Using the model in Eqs (3)–(6) we investigate different scenarios demonstrating the interactions between mechanics and signalling processes. First, we consider the impact of the cell Young's modulus $E_c$ and compare the dynamics when considering a constant $E_c$ versus the case where $E_c$ depends on activated FAK as defined in Eq (2). We also model the effect of the stress on the signalling molecules FAK and simulate equations Eq (6) for $C_1 = 0$ $(\text{kPa s})^{-1}$ and $C_1 = 0.1$ $(\text{kPa s})^{-1}$, respectively. Additionally we consider two experimental scenarios: (i) the cell is placed on a rigid substrate, modelled by the boundary conditions Eq (4) on $\Gamma \setminus \Gamma_0$ and Eq (5) on $\Gamma_0$ or (ii) the cell is embedded in an agar substrate and we apply the force boundary condition Eq (4) on the entire cell membrane. We also distinguish between two different stimuli, similar to [18], (i) the so called '2xD stimulus', where the substrate stiffness is only applied to the bottom of the cell, i.e. $E$ is nonzero only on $\Gamma_0$, and (ii) the '3D stimulus' where the cell is embedded in an agar (substrate) and the impact of the substrate stiffness on the signalling processes is considered on the whole cell membrane. To analyse the impact of the cell shape on the dynamics of signalling molecules and mechanical deformations we consider both axisymmetric cells and polarised cells with a lamellipodium like shape. The diameter of the cell is larger for the lamellipodium cells such that the volume is similar to the axisymmetric cells.

The parameters are chosen as in Table 1. For numerical simulations, we use a Finite Element Method to discretize in space and a semi-implicit Euler method to discretize in time,

**Table 1. Parameter values for simulations of the model in Eqs (3)–(6).**

(a) Parameters inherited from the model in Eq (1) that are identical to [18]. Because the goal is to compare and extend upon the model of [18], we choose the values to be the same as [18], which are based on literature and fitting to data. The exception are $D_1$ and $D_2$, which is discussed in S1 Appendix, Sect A.1.

| Parameters | | Value |
|---|---|---|
| $\phi_d^0$ | | 0.7 $\mu$mol/dm$^3$ |
| $\phi_a^0$ | | 0.3 $\mu$mol/dm$^3$ |
| $\rho_a^0$ | | $6 \cdot 10^{-7}$ $\mu$mol/dm$^2$ |
| $\rho_d^0$ | | 1 $\mu$mol/dm$^3$ |
| $D_1$ | | 4 $\mu$m$^2$/s |
| $D_2$ | | 4 $\mu$m$^2$/s |
| $D_3$ | | 0.3 $\mu$m$^2$/s |
| $k_1$ | | 0.035 s$^{-1}$ |
| $k_2$ | | 0.015 s$^{-1}$ |
| $k_3$ | | 0.379 s$^{-1}$ |
| $k_4$ | | 0.625 s$^{-1}$ |
| $k_5$ | | 0.0168 s$^{-1}$ |
| $E$ | | $0.1, 5.7, 7 \cdot 10^6$ kPa |
| $C$ | | 3.25 kPa |
| $n$ | | 5 |
| $\gamma$ | | 8.8068 dm$^3$/$\mu$mol |
| axisymmetric shape | $|Y|$ | 1193 $\mu$m$^3$ |
| | $|\Gamma|$ | 1020 $\mu$m$^2$ |
| lamellipodium shape | $|Y|$ | 1099 $\mu$m$^3$ |
| | $|\Gamma|$ | 1115 $\mu$m$^2$ |

(b) Parameter values for parameters introduced in this paper. A brief robustness analysis is performed on all parameters introduced in the coupled model in Eqs (3)–(6), see S1 Appendix, Sect A.6.

| Parameters | Value | Reference/Justification |
|---|---|---|
| $C_1$ | 0.1 (kPa s)$^{-1}$ | range 0–2 (kPa s)$^{-1}$ is explored in results |
| $k_6$ | 0.1 s$^{-1}$ | fitted to yield magnitude of deformation range $0 - 10$ $\mu$m [18] |
| $k_7$ | 0.2 kPa | fitted to results for $\phi_a$ in [18] |
| $k_8$ | 2.4245 dm$^3$/$\mu$mol | fitted to results for $\phi_a$ in [18] |
| $p$ | 2.6 | fitted to results for $\phi_a$ and F-actin in [18] and [38] |
| $\nu_c$ | 0.3 | estimated 0.17–0.66 [42] |

with the mesh size $h$ = 2.94 $\mu$m and time step $\Delta t$ = 0.5 s. Details on the numerical scheme and benchmark computations demonstrating the accuracy of the approach for a problem with a known solution are given in S1 Appendix, Sect A.3.

## Results

### Numerical simulations with 2xD stimulus

First we look at the results that would most reflect a cell on a substrate in vitro. Here, the substrate stiffness appears as a stimulus only on the bottom boundary of the cell, i.e. $E$ is nonzero only on $\Gamma_0$, and deformation is restricted in the vertical direction at the bottom boundary of the cell. The results for the axisymmetric shape of the cell are found in Fig 2, whereas results for the lamellipodium shape are presented in Fig 3. Note that results for $\phi_a$ and $\rho_a$ when $E_c$ = 0.6 kPa and $C_1 = 0$ (kPa s)$^{-1}$ are identical to the one without mechanics in S1 Appendix, Sect A.1. In this case, we see that the magnitude of the deformation $|u|$ is largest at the edge of the cell. The cell expands axisymmetrically at the base. As expected, the expansion is larger for higher concentrations of $\phi_a$. For a lower substrate stiffness, $E$ = 0.1 kPa, the cell barely expands. When $C_1 = 0.1$ (kPa s)$^{-1}$, the concentrations of $\phi_a$ and $\rho_a$ and the magnitude of the deformation $|u|$ increase, with a bigger increase for lower substrate stiffness and a smaller

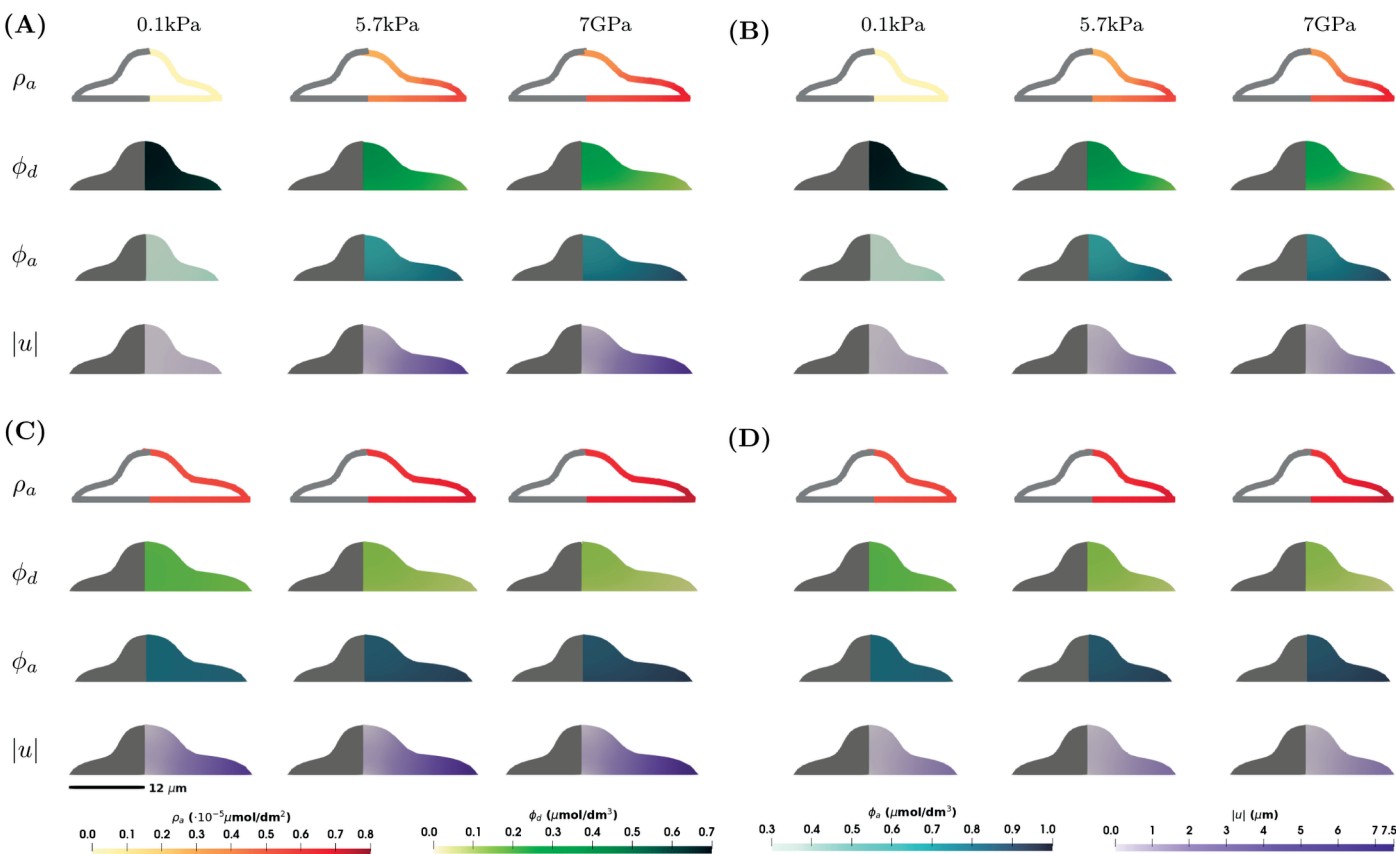

**Fig 2. Numerical simulation results showing $\rho_a$, $\phi_d$, $\phi_a$ and $|u|$ for the model in Eqs (3)–(6) for the axisymmetric shape and in the case of 2xD stimulus at a steady state at** $T = 100$ s. Four different scenarios are considered: **(A)** $C_1 = 0$ $(\text{kPa s})^{-1}$ $(\sigma \nrightarrow \phi_a)$ and $E_c = 0.6$ kPa $(\phi_a \nrightarrow E_c)$; **(B)** $C_1 = 0$ $(\text{kPa s})^{-1}$ $(\sigma \nrightarrow \phi_a)$ and $E_c = f(\phi_a)$ $(\phi_a \to E_c)$; **(C)** $C_1 = 0.1$ $(\text{kPa s})^{-1}$ $(\sigma \to \phi_a)$ and $E_c = 0.6$ kPa $(\phi_a \nrightarrow E_c)$; **(D)** $C_1 = 0.1$ $(\text{kPa s})^{-1}$ $(\sigma \to \phi_a)$ and $E_c = f(\phi_a)$ $(\phi_a \to E_c)$. Within each subfigure, the rows represent $\rho_a$, $\phi_d$, $\phi_a$ and $|u|$ on a cross-section of the plane $x_1 = 0$ of the axisymmetric cell, and the columns represent $E = 0.1, 5.7, 7 \cdot 10^6$ kPa. Parameter values as in Table 1.

increase for larger substrate stiffness. When comparing $E_c = 0.6$ kPa and $E_c = f(\phi_a)$, the deformations show similar patterns, expanding at the base of the cell, however, the magnitude of the deformation is much lower in the case $E_c = f(\phi_a)$. This is probably because $E_c = f(\phi_a) \approx 0.6$ kPa for a small substrate stiffness $E$, but is doubled in magnitude for larger substrate stiffness, see Fig 4. The larger cell Young's modulus $E_c$ means it is harder for the cell to deform, resulting in a lower magnitude of deformation. This difference illustrates that, unlike the constant Young's modulus case, a concentration-dependent Young's modulus allows for potential homeostasis and adaptation of cell mechanics to different values of the substrate stiffness [28].

For the two-way couplings between the mechanics and chemistry, i.e. $E_c = f(\phi_a)$ and $C_1 = 0.1$ $(\text{kPa s})^{-1}$, we see similar results for the deformation as when $E_c = f(\phi_a)$ and $C_1 = 0$ $(\text{kPa s})^{-1}$. The main difference is that the deformation for $E = 0.1$ kPa is now at a similar magnitude as for the larger substrate stiffnesses, demonstrating the importance of the signalling processes in the adaptation of cell mechanics to changing environmental conditions.

Comparing the simulation results for the two different shapes in Figs 2 and 3, the concentration of activated RhoA, $\rho_a$, is slightly lower for the lamellipodium shape. For the lamellipodium shape, we observe the largest deformations at the corners furthest from the nucleus.

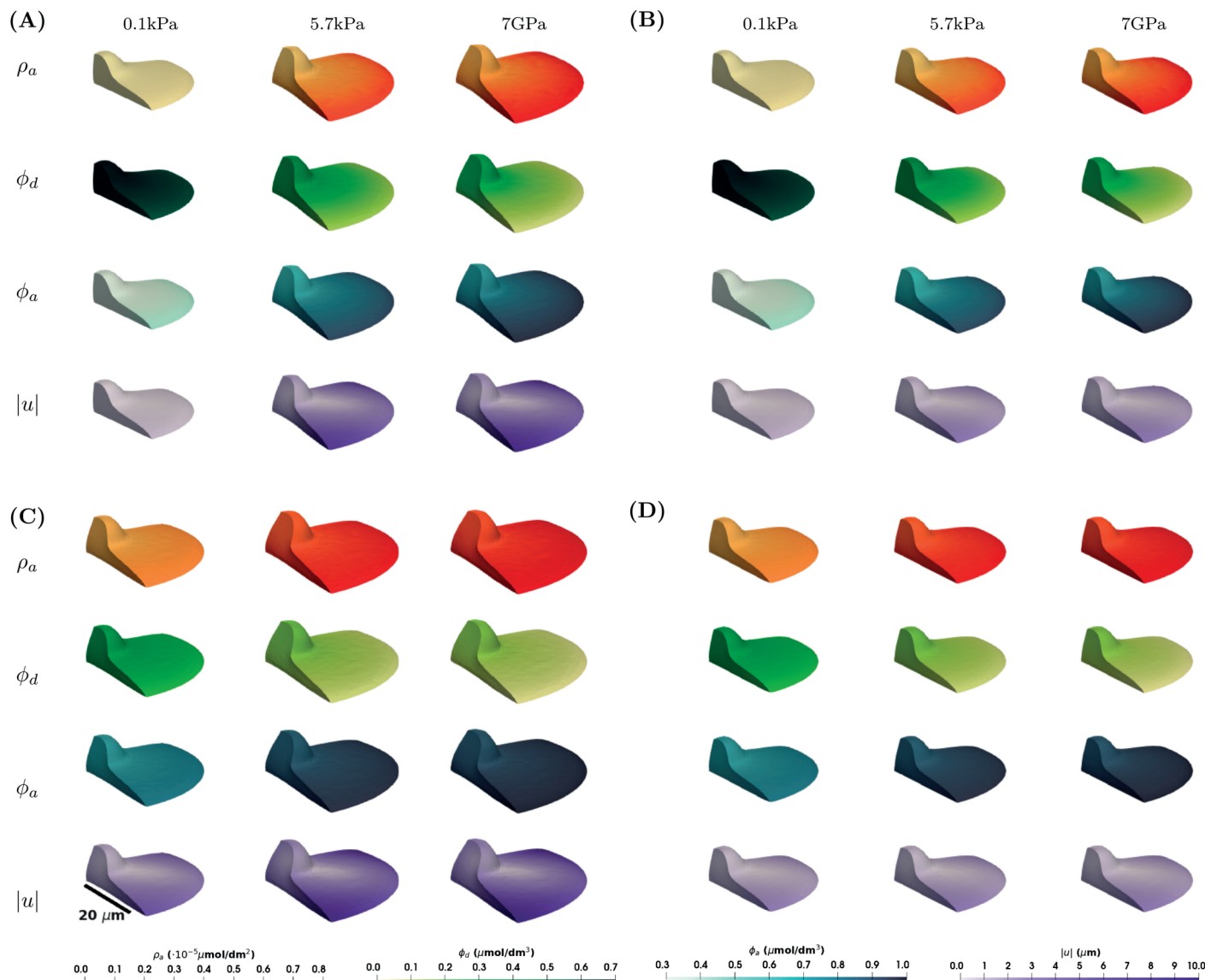

**Fig 3. Numerical simulation results showing $\rho_a$, $\phi_d$, $\phi_a$ and $|u|$ for the model in Eqs (3)–(6) for the lamellipodium shape and in the case of 2xD stimulus at a steady state at** $T = 100$ **s.** Four different scenarios are considered: **(A)** $C_1 = 0$ (kPa s)$^{-1}$ ($\sigma \nrightarrow \phi_a$) and $E_c = 0.6$ kPa ($\phi_a \nrightarrow E_c$); **(B)** $C_1 = 0$ (kPa s)$^{-1}$ ($\sigma \nrightarrow \phi_a$) and $E_c = f(\phi_a)$ ($\phi_a \rightarrow E_c$); **(C)** $C_1 = 0.1$ (kPa s)$^{-1}$ ($\sigma \rightarrow \phi_a$) and $E_c = 0.6$ kPa ($\phi_a \nrightarrow E_c$); **(D)** $C_1 = 0.1$ (kPa s)$^{-1}$ ($\sigma \rightarrow \phi_a$) and $E_c = f(\phi_a)$ ($\phi_a \rightarrow E_c$). Within each subfigure, the rows represent $\rho_a$, $\phi_d$, $\phi_a$ and $|u|$ on the surface of the cell, and the columns represent $E = 0.1, 5.7, 7 \cdot 10^6$ kPa. Parameter values as in Table 1.

Fig 4 summarises the results at time $T = 100$ s by plotting the mean, $\frac{1}{|\Omega|} \int_\Omega \cdot \mathrm{d}x$, of $E_c = f(\phi_a)$, the volume change $\mathrm{div}(u)$, $\phi_a$, and $\rho_a$ as functions of the substrate stiffness $E$, with the bars being the range of these variables, for different values of the constant $C_1$ in the activation of FAK by the cell stress. As expected, an increase in $C_1$ results in an increase in the concentration of activated FAK, $\phi_a$. The dependence of $\phi_a$ on the substrate stiffness $E$, especially for $C_1 = 0$ (kPa s)$^{-1}$ and $C_1 = 0.5$ (kPa s)$^{-1}$, resembles a Hill function representing a threshold response. This agrees with simulations in [18] which themselves fit experimental observations presented in [44]. For most of the cases, the results for the lamellipodium shape are very similar to the results for the axisymmetric shape. However, for $C_1 = 0.1$ (kPa s)$^{-1}$, the magnitude

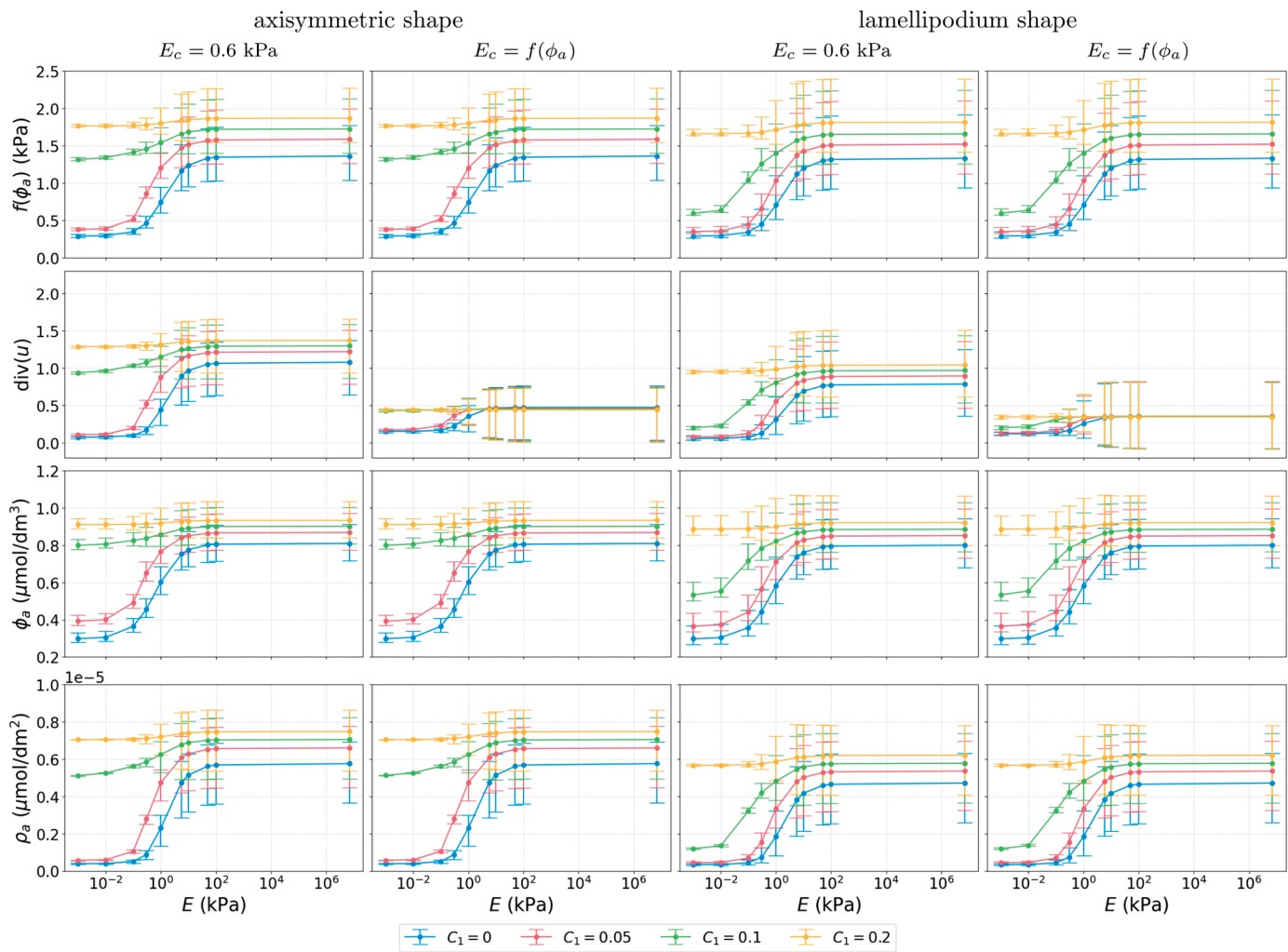

**Fig 4. Simulation results showing the mean, $\frac{1}{|\Omega|}\int_\Omega \cdot dx$, min and max values of $f(\phi_a)$, $\text{div}(u)$, $\phi_a$ and $\rho_a$ as functions of substrate stiffness $E$.** We consider different couplings with four different values for $C_1$ and two different shapes at $T = 100$ s by which time the results are at a steady state. All other parameter values as in Table 1.

of the threshold-like response in all variables is bigger in the lamellipodium case. In terms of the Young's modulus, when $E_c = 0.6$ kPa we observe much larger volume changes than when $E_c = f(\phi_a)$ in all the numerical experiments.

## Numerical simulations for the 3D stimulus case on a rigid substrate

In numerical simulations for a 3D stimulus on a rigid substrate, the substrate stiffness affects the whole cell membrane and we consider the boundary conditions Eq (4) on $\Gamma \setminus \Gamma_0$ and Eq (5) on $\Gamma_0$. The results for numerical experiments can be found in Figs 5 and 6. Overall, the concentrations $\phi_a$ and $\rho_a$ are larger than in the case of the 2xD stimulus, which is in line with the results in [18]. The higher concentrations of $\rho_a$ results in larger deformations, where the maximum magnitude of the deformation in the case of the 2xD stimulus was 7 $\mu$m, see Fig 2, while the maximum magnitude of the deformation in the case of the 3D stimulus is 7.5 $\mu$m, see Fig 2. Similar behaviour is observed for the lamellipodium shape, see Figs 3 and

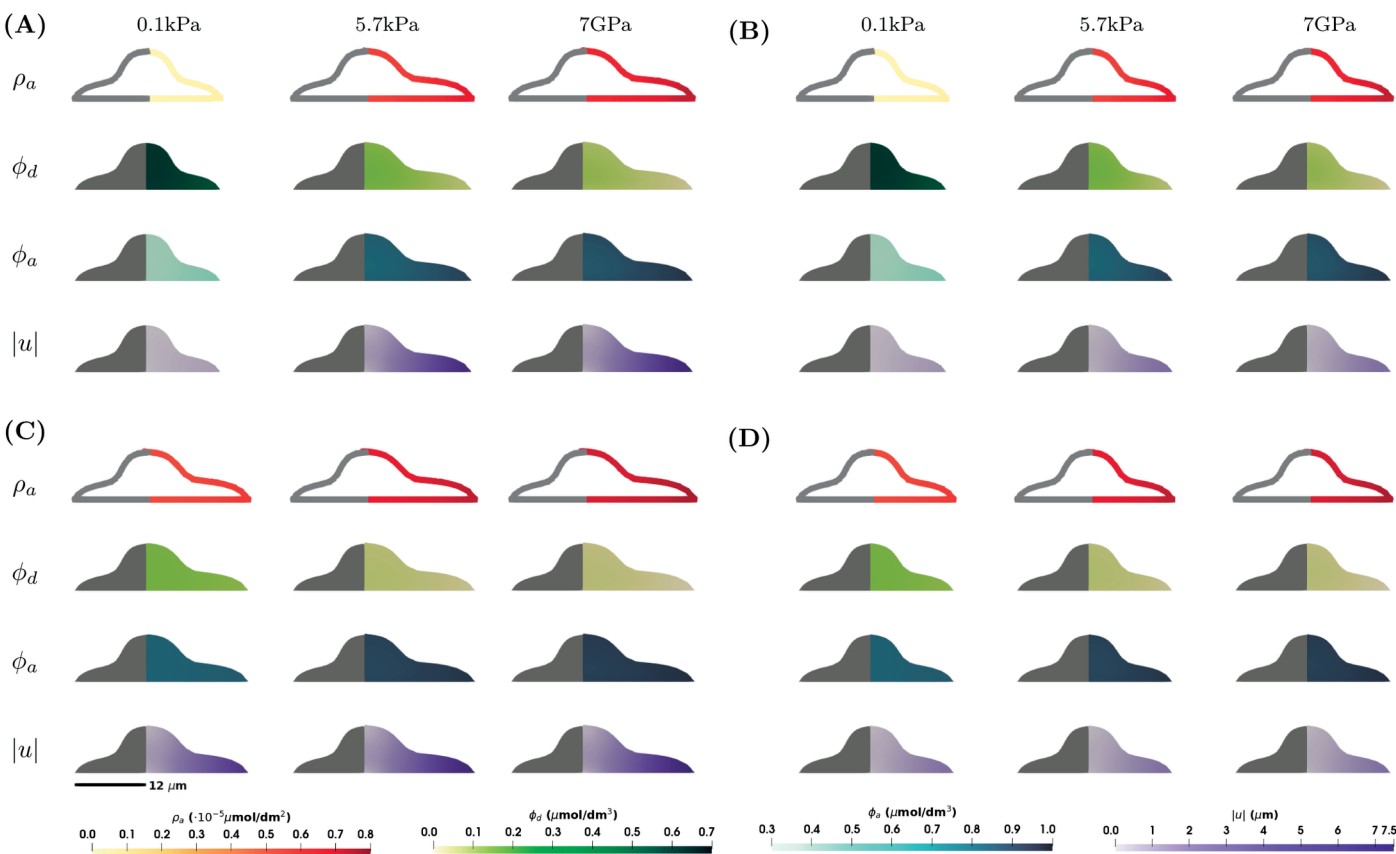

**Fig 5. Numerical simulation results showing $\rho_a$, $\phi_d$, $\phi_a$ and $|u|$ for the model in Eqs (3)–(6) for the axisymmetric shape and in the case of the 3D stimulus at a steady state at** $T = 100$ s. Four different scenarios are considered: **(A)** $C_1 = 0$ (kPa s)$^{-1}$ ($\sigma \nrightarrow \phi_a$) and $E_c = 0.6$ kPa ($\phi_a \nrightarrow E_c$); **(B)** $C_1 = 0$ (kPa s)$^{-1}$ ($\sigma \nrightarrow \phi_a$) and $E_c = f(\phi_a)$ ($\phi_a \to E_c$); **(C)** $C_1 = 0.1$ (kPa s)$^{-1}$ ($\sigma \to \phi_a$) and $E_c = 0.6$ kPa ($\phi_a \nrightarrow E_c$); **(D)** $C_1 = 0.1$ (kPa s)$^{-1}$ ($\sigma \to \phi_a$) and $E_c = f(\phi_a)$ ($\phi_a \to E_c$). Within each subfigure, the rows represent $\rho_a$, $\phi_d$, $\phi_a$ and $|u|$ on a cross-section of the plane $x_1 = 0$ of the axisymmetric cell, and the columns represent $E = 0.1, 5.7, 7 \cdot 10^6$ kPa. Parameter values as in Table 1.

S1. Another difference between two cases are larger variations in concentration and a larger difference between maximal and minimal values in the case of the 2xD stimulus than in the case of 3D stimulus, see Figs 4 and 6. Similar behaviour is observed also in the model for the signalling processes without mechanics, see S1 Appendix, Sect A.1.

## Numerical simulations for the model in Eqs (3), (4), and (6).

To investigate a setting more close to a cell in vivo, we consider the coupled model in Eqs (3), (4), (6) with force boundary conditions on the whole cell membrane, without restricting the deformation on the bottom of the cell.

**Numerical simulations in the case of 3D stimulus.** Simulation results for a 3D stimulus that models a cell surrounded by the extracellular matrix are presented in Figs 7 and 8. Corresponding results of the evolution of the mean of $f(\phi_a)$, div($u$), $\phi_a$ and $\rho_a$ can be found in S1 Appendix, Sect A.2. The results show the same differences between the different couplings as in Figs 5 and 6. Comparing Figs 5 and 7, the results for the concentrations $\phi_a$ and $\rho_a$ are indistinguishable, however there is a clear difference in deformation of the bottom

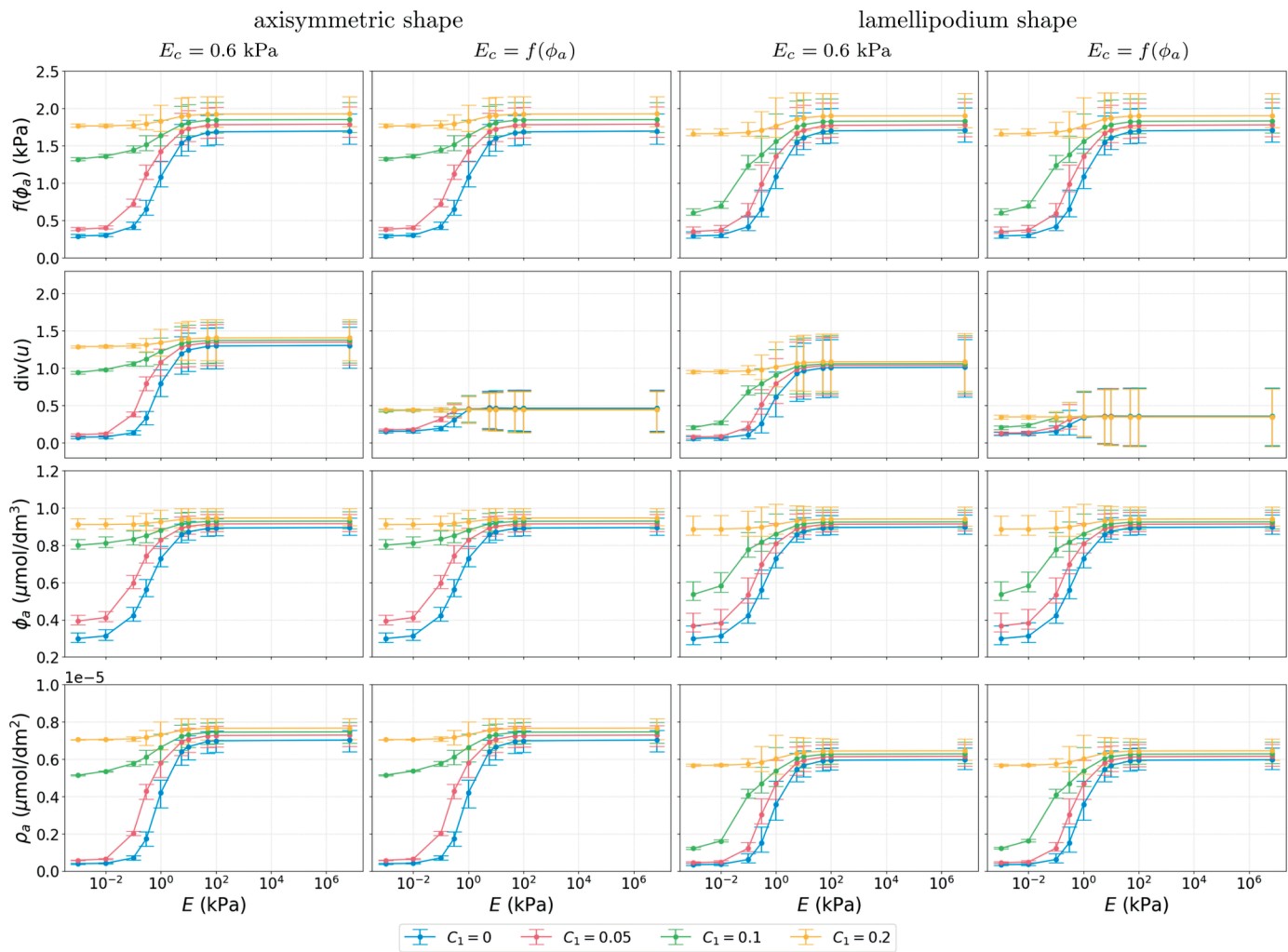

**Fig 6. Simulation results showing the mean, $\frac{1}{|\Omega|}\int_\Omega \cdot dx$, min and max values of $f(\phi_a)$, $\mathrm{div}(u)$, $\phi_a$ and $\rho_a$ as functions of substrate stiffness $E$.** We consider different couplings with four different values for $C_1$ and two different shapes at $T = 100$ s by which time the results are at a steady state. All other parameter values as in Table 1.

of the cell and in the case of the fixed vertical deformations the magnitude of the deformation at the base of the cell is slightly lower than in the case of force boundary conditions. The same differences are observed for the lamellipodium shape case, see S1 and S2 Figs. The concentrations of the signalling molecules are also less sensitive to parameter changes than the volume change, as can be seen from the parameter analysis in S1 Appendix, Sect A.6. Comparing Figs 6 and 8, the main difference is in the behaviour of $\mathrm{div}(u)$ as a function of $E$. Even though the average volume change is the same, we see differences in the maximum and minimum values of the local volume change across the domain. In particular, the maximum local volume change when considering the model with a partially fixed boundary is larger and is located on the base of the cell, while the maximum local volume change when considering the model with the force boundary conditions is smaller, but the cell deforms more evenly in all directions.

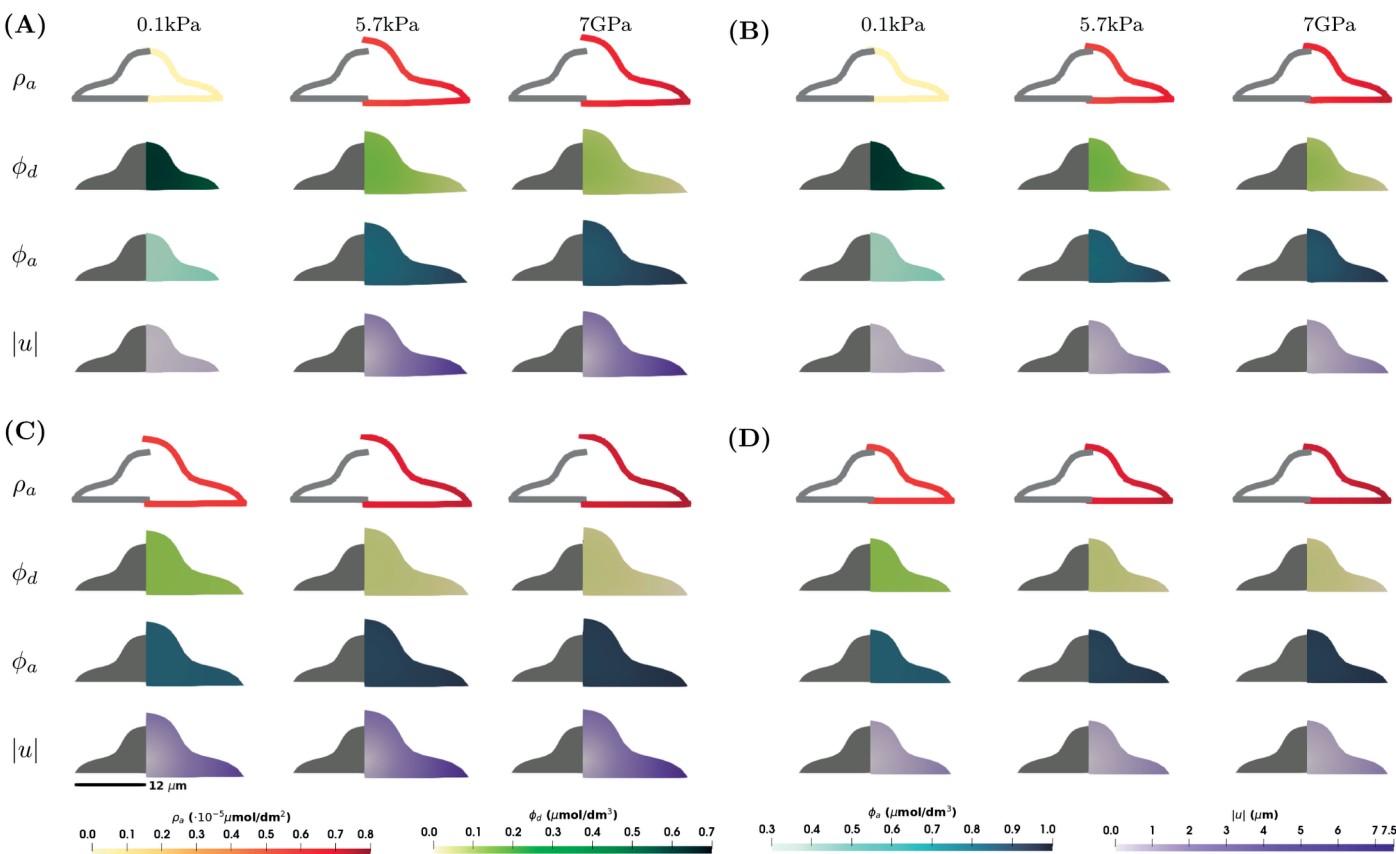

**Fig 7. Numerical simulation results showing $\rho_a$, $\phi_d$, $\phi_a$ and $|u|$ for the model in Eqs (3), (4), and (6) for the axisymmetric shape and in the case of the 3D stimulus at a steady state at $T = 100$ s.** Four different scenarios are considered: **(A)** $C_1 = 0$ (kPa s)$^{-1}$ ($\sigma \nrightarrow \phi_a$) and $E_c = 0.6$ kPa ($\phi_a \nrightarrow E_c$); **(B)** $C_1 = 0$ (kPa s)$^{-1}$ ($\sigma \nrightarrow \phi_a$) and $E_c = f(\phi_a)$ ($\phi_a \rightarrow E_c$); **(C)** $C_1 = 0.1$ (kPa s)$^{-1}$ ($\sigma \rightarrow \phi_a$) and $E_c = 0.6$ kPa ($\phi_a \nrightarrow E_c$); **(D)** $C_1 = 0.1$ (kPa s)$^{-1}$ ($\sigma \rightarrow \phi_a$) and $E_c = f(\phi_a)$ ($\phi_a \rightarrow E_c$). Within each subfigure, the rows represent $\rho_a$, $\phi_d$, $\phi_a$ and $|u|$ on a cross-section of the plane $x_1 = 0$ of the axisymmetric cell, and the columns represent $E = 0.1, 5.7, 7 \cdot 10^6$ kPa. Parameter values as in Table 1.

**Numerical simulations in the case of 2xD stimulus.** In Figs 9, S3 and S4 we report on simulation results in the case of 2xD stimulus and force boundary conditions applied to the entire boundary. For the concentrations, the results are similar to the results in the case of 2xD stimulus and no vertical deformation on the bottom of the cell, see Figs 2 and 9. However, the results for the deformation are different compared to the previous results. In Fig 9, the cell does not just expand but changes shape as the edges of the cell deform upwards, which is not possible in the case of the partially fixed boundary as we assume no vertical deformation at the base. The deformation of the cell upwards can also be observed in the case of the 3D stimulus, but it is smaller due to the impact of the ECM surrounding the cell, see Fig 7. We observe that for $C_1 = 0$ (kPa s)$^{-1}$ the cell deforms upwards a little more than for $C_1 = 0.1$ (kPa s)$^{-1}$. This is due to the larger variation in the concentration $\rho_a$ for $C_1 = 0$ (kPa s)$^{-1}$ compared to $C_1 = 0.1$ (kPa s)$^{-1}$. The same features are observed for the lamellipodium shape, see Figs 3, S2, S3 and S4.

## Discussion and conclusion

We have derived a model for mechanotransduction via the RhoA signalling pathway with ECM stiffness and intracellular mechanical properties serving as the mechanical cues. The

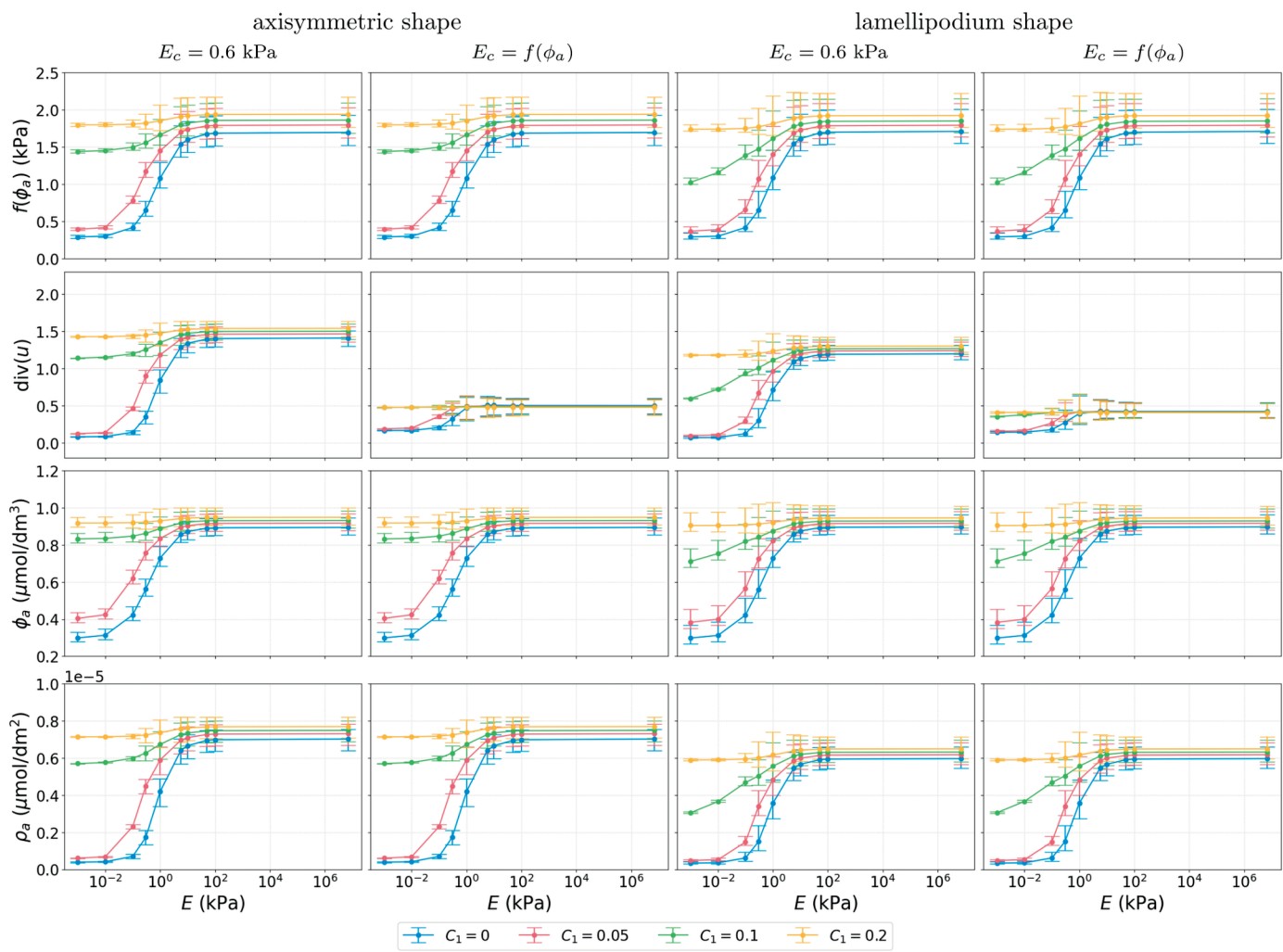

**Fig 8. Simulation results showing the mean, $\frac{1}{|\Omega|}\int_{\Omega}\cdot dx$, min and max values of $f(\phi_a)$, div$(u)$, $\phi_a$ and $\rho_a$ as functions of substrate stiffness $E$, in the case of the model in Eqs (3), (4) and (6) and 3D stimulus.** We consider different couplings, four different values for $C_1$, and two different shapes at $T = 100$ s by which time the results are at a steady state. All other parameter values as in Table 1.

modelling extends the work of [18] incorporating the explicit modelling of cell deformation based on an elastic constitutive assumption. We have extended on [18,23,24] and introduced a two-way coupling between the mechanics of the cell and biochemical signalling processes. This two-way coupling appears to be central to mechanical homeostasis observed in biological experiments [28]. We propose a robust numerical method, based on the bulk-surface finite element method (FEM), see e.g. [26], for the approximation of the model and report on simulation results for different scenarios, validating the results by comparison with simulations presented in [18] and experimental observations in [44]. Namely, we considered different levels of substrate stiffness for cells of different shapes that either sit on a rigid flat substrate or are embedded in a three-dimensional substrate.

Our broad conclusions are that cell shape strongly influences the dynamics of the signalling molecules and the deformation of the cell, as seen in all figures comparing the axisymmetric and lamellipodium shape, where the emergent patterns differ, which is in line with

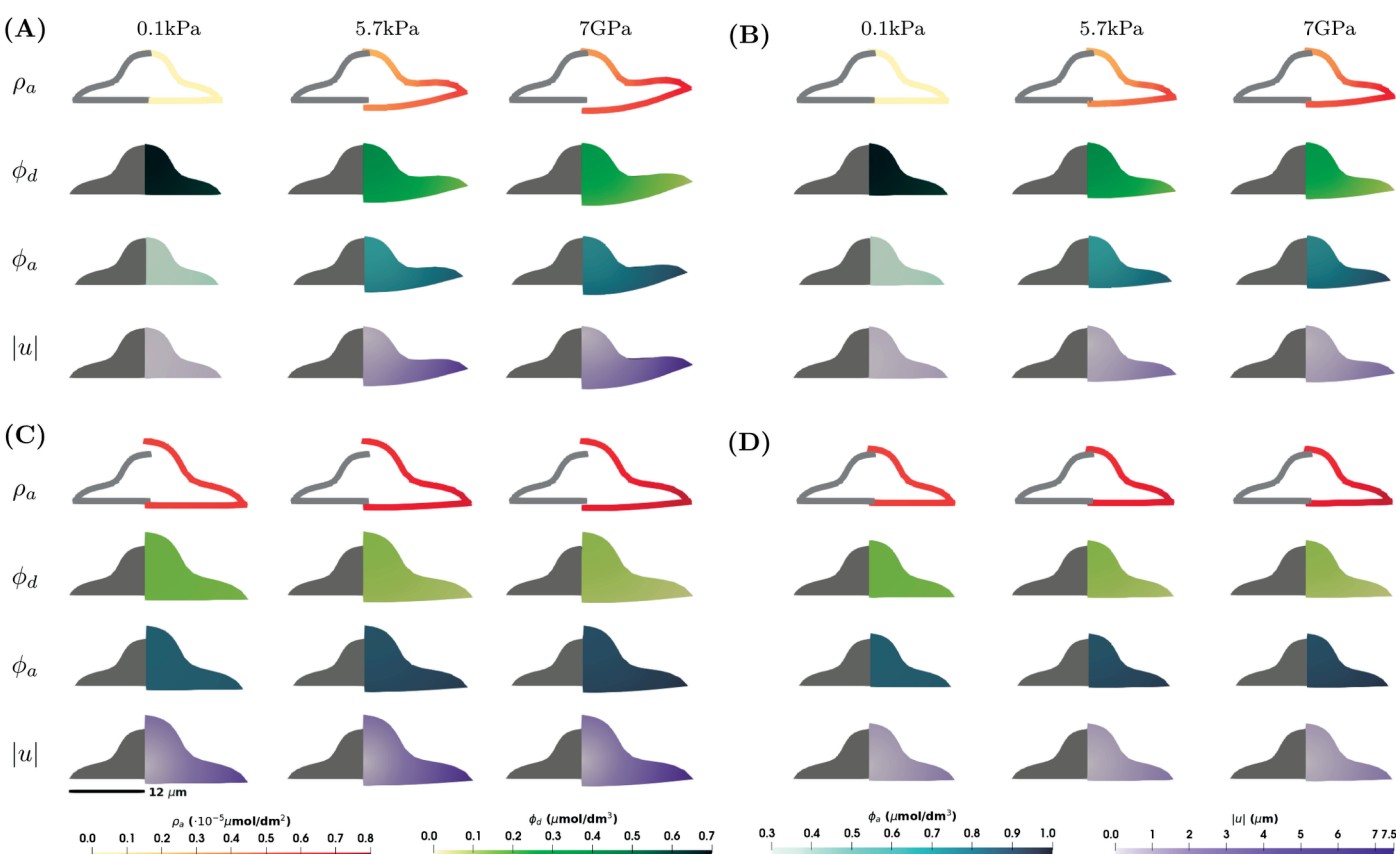

**Fig 9. Numerical simulation results showing $\rho_a$, $\phi_d$, $\phi_a$ and $|u|$ simulation results showing $\phi_a$ and $\rho_a$ for the model in Eqs (3), (4), and (6) for the axisymmetric shape and in the case of 2xD stimulus at a steady state at** $T = 100$ **s.** Four different scenarios are considered: **(A)** $C_1 = 0$ (kPa s)$^{-1}$ ($\sigma \nrightarrow \phi_a$) and $E_c = 0.6$ kPa ($\phi_a \nrightarrow E_c$); **(B)** $C_1 = 0$ (kPa s)$^{-1}$ ($\sigma \nrightarrow \phi_a$) and $E_c = f(\phi_a)$ ($\phi_a \rightarrow E_c$); **(C)** $C_1 = 0.1$ (kPa s)$^{-1}$ ($\sigma \rightarrow \phi_a$) and $E_c = 0.6$ kPa ($\phi_a \nrightarrow E_c$); **(D)** $C_1 = 0.1$ (kPa s)$^{-1}$ ($\sigma \rightarrow \phi_a$) and $E_c = f(\phi_a)$ ($\phi_a \rightarrow E_c$). Within each subfigure, the rows represent $\rho_a$, $\phi_d$, $\phi_a$ and $|u|$ on a cross-section of the plane $x_1 = 0$ of the axisymmetric cell, and the columns represent $E = 0.1, 5.7, 7 \cdot 10^6$ kPa. Parameter values as in Table 1.

experimental observations [45,46]. Cell shape also affects experimentally observed features such as the threshold-like response to changes in substrate stiffness [44] which is reproduced by the model. In Figs 4, 6, 8 and S4, we see that for certain parameters ($C_1 = 0.1$ (kPa s)$^{-1}$ and low substrate stiffness), the cell shape affects the mean concentrations of the signalling molecules and the mean volume change of the cell, and thus changes the threshold-like response.

Our simulations exhibit novel emergent features, that are inaccessible without the framework we propose, such as the bidirectional coupling between mechanics and signalling processes through allowing the Young's modulus of the cell to depend on protein concentration that can allow for robustness in terms of the magnitude of deformation in response to differences in substrate stiffness. This is an example of a mechanical homeostasis mechanism that emerges only at this level of modelling complexity which is of relevance to biology [28]. Other instances of mechanical homeostasis are the stress being maintained in the cardiovascular system under mechanical perturbations [47] and the tensional homeostasis by the RhoA signalling pathway at the level of multiple cells [48,49], which is known to be governed by cellular stiffness sensing [50]. Another mechanism that experiences homeostatic response to substrate stiffness is that of the mechanical memory of the cell, describing the phenomenon of a

cell responding less to substrates with lower stiffness if they have been cultured on stiff substrates [49,51]. Due to the bidirectional coupling between the mechanics and the chemistry in our modelling framework, an extension of this work by changing the chosen couplings could be used to model these other mechanical homeostasis phenomena.

Based on previous biological studies [18,38], we considered cases in which the mechanical properties of the cell (cell stiffness) depend on the concentration of signalling molecules. This coupling yields less sensitivity of total deformation to substrate stiffness whilst leaving the dynamics of the signalling molecules themselves broadly unchanged, see Figs 4, 6, 8 and S4. The insensitivity of the dynamics of the signalling molecules to deformation levels arises since in the model proposed here they are influenced by the local stress rather than deformation. We note that the above constitutes another emergent homeostasis mechanism that the modelling framework allows us to explore. We stress that our work serves as an example of how mechanotransduction may be modelled and more complicated models for the mechanics, biochemistry and couplings therefore are warranted based on the remarkable emergent features we observe even in our relatively simple setting. We expect such models to be particularly fruitful avenues for future work.

One such example of more complicated models for the mechanics could include a viscoelastic or poroelastic constitutive law. As presented in Sect A.7 in S1 Appendix, the assumption of a (linear) viscoelastic constitutive law leads to qualitatively similar results to those presented in this work for a purely (linear) elastic constitutive law. Our current assumption of a linear elastic constitutive law for the mechanics of the cell is limiting as it assumes small deformations. This framework needs to be extended to study the effect of large deformations and shape changes, which would include the effect these deformations have on the signalling molecules. This would be especially interesting, as this study shows that cell shape is one of the determinants affecting the mean concentrations of the signalling molecules.

The boundary conditions for the deformation we consider correspond to simple models of a cell in vitro (flat 2D substrate) or in vivo (homogeneous 3D matrix). We see that the cell on a 2D substrate appears to spread radially with minimal deformation orthogonal to the substrate while the latter exhibits a more uniform although smaller in total magnitude 3D deformation. Differences in deformation for different environments are in line with the literature as the effect of the substrate stiffness on cells is known to vary in 2D and 3D substrates [52]. An interesting extension that could be included in the above framework would be spatial variations in substrate stiffness or more complicated models for the substrate mechanics both of which are of much biological relevance [53–55].

This work shows how mechanistic modelling of mechanotransduction can reveal remarkable emergent properties. It lays the groundwork for future studies where further complexity can be added as required to model specific signalling pathways or to reflect other mechanical models derived from different constitutive assumptions. We anticipate that choosing a viscoelastic or poroelastic constitutive law for the mechanics of the cell is an interesting direction for future studies, as this is in line with recent experimental observations [31,32]. Given the fact that cell shape greatly influences the dynamics of the cell, as shown in this work, other reference geometries are also of interest as a subject for future work. Extending the signalling model of [18] further, we intend to couple the model of this work with a similar biomechanical model for the deformation of the nucleus coupled with the dynamics of signalling molecules within the nucleus, such as the YAP/TAZ pathway [56].

## Supporting information

**S1 Appendix. Numerical scheme and simulations.** The appendix contains a comparison of the reduced model and the full model of [18] in Sect A.1, temporal statistics in Sect A.2, the numerical scheme in Sect A.3, the conversion from $\mu M$ to $\#/\mu m^2$ in Sect A.4, simulations for the model with nucleus in Sect A.5, a parameter sensitivity analysis in Sect A.6, and simulations for viscoelastic model in Sect A.7.
(PDF)

**S1 Fig. Numerical simulation results showing $\rho_a$, $\phi_d$, $\phi_a$ and $|u|$ for the model in Eqs (3)–(6) for the lamellipodium shape and in the case of the 3D stimulus at a steady state at $T$ = 100 s.** Four different scenarios are considered: **(A)** $C_1 = 0$ (kPa s)$^{-1}$ ($\sigma \nrightarrow \phi_a$) and $E_c = 0.6$ kPa ($\phi_a \nrightarrow E_c$); **(B)** $C_1 = 0$ (kPa s)$^{-1}$ ($\sigma \nrightarrow \phi_a$) and $E_c = f(\phi_a)$ ($\phi_a \rightarrow E_c$); **(C)** $C_1 = 0.1$ (kPa s)$^{-1}$ ($\sigma \rightarrow \phi_a$) and $E_c = 0.6$ kPa ($\phi_a \nrightarrow E_c$); **(D)** $C_1 = 0.1$ (kPa s)$^{-1}$ ($\sigma \rightarrow \phi_a$) and $E_c = f(\phi_a)$ ($\phi_a \rightarrow E_c$). Within each subfigure, the rows represent $\rho_a$, $\phi_d$, $\phi_a$ and $|u|$ on the surface of the cell, and the columns represent $E = 0.1, 5.7, 7 \cdot 10^6$ kPa. Parameter values as in Table 1.
(TIF)

**S2 Fig. Numerical simulation results showing $\rho_a$, $\phi_d$, $\phi_a$ and $|u|$ for the model in Eqs (3), (4), and (6) for the lamellipodium shape and in the case of the 3D stimulus at a steady state at $T$ = 100 s.** Four different scenarios are considered: **(A)** $C_1 = 0$ (kPa s)$^{-1}$ ($\sigma \nrightarrow \phi_a$) and $E_c = 0.6$ kPa ($\phi_a \nrightarrow E_c$); **(B)** $C_1 = 0$ (kPa s)$^{-1}$ ($\sigma \nrightarrow \phi_a$) and $E_c = f(\phi_a)$ ($\phi_a \rightarrow E_c$); **(C)** $C_1 = 0.1$ (kPa s)$^{-1}$ ($\sigma \rightarrow \phi_a$) and $E_c = 0.6$ kPa ($\phi_a \nrightarrow E_c$); **(D)** $C_1 = 0.1$ (kPa s)$^{-1}$ ($\sigma \rightarrow \phi_a$) and $E_c = f(\phi_a)$ ($\phi_a \rightarrow E_c$). Within each subfigure, the rows represent $\rho_a$, $\phi_d$, $\phi_a$ and $|u|$ on the surface of the cell, and the columns represent $E = 0.1, 5.7, 7 \cdot 10^6$ kPa. Parameter values as in Table 1.
(TIF)

**S3 Fig. Numerical simulation results showing $\rho_a$, $\phi_d$, $\phi_a$ and $|u|$ for the model in Eqs (3), (4), and (6) for the lamellipodium shape and in the case of 2xD stimulus at a steady state at $T$ = 100 s.** Four different scenarios are considered: **(A)** $C_1 = 0$ (kPa s)$^{-1}$ ($\sigma \nrightarrow \phi_a$) and $E_c = 0.6$ kPa ($\phi_a \nrightarrow E_c$); **(B)** $C_1 = 0$ (kPa s)$^{-1}$ ($\sigma \nrightarrow \phi_a$) and $E_c = f(\phi_a)$ ($\phi_a \rightarrow E_c$); **(C)** $C_1 = 0.1$ (kPa s)$^{-1}$ ($\sigma \rightarrow \phi_a$) and $E_c = 0.6$ kPa ($\phi_a \nrightarrow E_c$); **(D)** $C_1 = 0.1$ (kPa s)$^{-1}$ ($\sigma \rightarrow \phi_a$) and $E_c = f(\phi_a)$ ($\phi_a \rightarrow E_c$). Within each subfigure, the rows represent $\rho_a$, $\phi_d$, $\phi_a$ and $|u|$ on the surface of the cell, and the columns represent $E = 0.1, 5.7, 7 \cdot 10^6$ kPa. Parameter values as in Table 1.
(TIF)

**S4 Fig. Simulation results showing the mean, $\frac{1}{|\Omega|} \int_\Omega \cdot \mathrm{d}x$, min and max values of $f(\phi_a)$, div$(u)$, $\phi_a$ and $\rho_a$ as functions of substrate stiffness $E$, in the case of the model in Eqs (3), (4) and (6) and 2xD stimulus.** We consider different couplings, four different values for $C_1$, and two different shapes at $T$ = 100 s by which time the results are at a steady state. All other parameter values as in Table 1.
(TIF)

## Acknowledgments

SV and MP would like to thank the Isaac Newton Institute for Mathematical Sciences, Cambridge, for support and hospitality during the research programme 'Uncertainty quantification and stochastic modelling of materials', EPSRC Grant Number EP/R014604/1, where some work on the manuscript was undertaken. The authors would like to thank Padmini Rangamani for helpful discussions.

## Author contributions

**Conceptualization:** Sofie Verhees, Chandrasekhar Venkataraman, Mariya Ptashnyk.

**Formal analysis:** Sofie Verhees, Chandrasekhar Venkataraman, Mariya Ptashnyk.

**Funding acquisition:** Mariya Ptashnyk.

**Methodology:** Sofie Verhees, Chandrasekhar Venkataraman, Mariya Ptashnyk.

**Software:** Sofie Verhees.

**Supervision:** Chandrasekhar Venkataraman, Mariya Ptashnyk.

**Visualization:** Sofie Verhees, Chandrasekhar Venkataraman, Mariya Ptashnyk.

**Writing – original draft:** Sofie Verhees, Chandrasekhar Venkataraman, Mariya Ptashnyk.

**Writing – review & editing:** Sofie Verhees, Chandrasekhar Venkataraman, Mariya Ptashnyk.

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
