## [Decision Letter · Decision Letter 0]

10 Mar 2025

PCOMPBIOL-D-24-02090

Mathematical modelling of mechanotransduction via RhoA signalling pathways

PLOS Computational Biology

Dear Dr. Verhees,

Thank you for submitting your manuscript to PLOS Computational Biology. After careful consideration, we feel that it has merit but does not fully meet PLOS Computational Biology's publication criteria as it currently stands. Therefore, we invite you to submit a revised version of the manuscript that addresses the points raised during the review process.

Please submit your revised manuscript within 60 days May 10 2025 11:59PM. If you will need more time than this to complete your revisions, please reply to this message or contact the journal office at ploscompbiol@plos.org. Please include the following items when submitting your revised manuscript:

We look forward to receiving your revised manuscript.

Kind regards,

Jochen Hub

Academic Editor

PLOS Computational Biology

Jason Papin

Editor-in-Chief

PLOS Computational Biology

**Additional Editor Comments (if provided):**

Both reviewers acknowledge the novelty and relevance of the study. However, they have identified weaknesses that require a thorough major revision, which may involve modifying the mathematical model and running new simulations.

The main manuscript contains an exceptionally large number of figures (20). Please consider whether key results can be presented more concisely in the main text and whether control simulations can be moved to the Supporting Information.

I look forward to your revised manuscript.

**Journal Requirements:**

**Reviewers' comments:**

Reviewer's Responses to Questions

**Comments to the Authors:**

Reviewer #1: This manuscript "Mathematical modelling of mechanotransduction via RhoA signalling pathways" presents a well-structured mathematical model describing the bidirectional coupling between RhoA signaling and cellular mechanics. Using a bulk-surface finite element method, the authors solve nonlinear reaction-diffusion equations alongside elasticity equations to simulate cellular deformations. Their findings highlight the influence of cell shape on RhoA dynamics and the role of mechanotransduction in maintaining mechanical homeostasis. While the study is novel and insightful, certain aspects require further refinement.

Detailed comments are listed below:

1. The authors assume that the cell follows a linear elasticity model, whereas many studies suggest that biological cells exhibit viscoelastic or poroelastic properties, especially over long time scales. It is recommended to discuss the validity of this assumption and consider incorporating viscoelastic effects to improve the biological realism of the model.

2. The model parameters (e.g., RhoA reaction rates, diffusion coefficients, substrate stiffness) are not well justified. The authors should provide references to experimental studies or parameter sensitivity analyses to strengthen the biological basis of their choices.

3. RhoA primarily regulates actin cytoskeleton remodeling, influencing cellular contractility and adhesion dynamics. However, the current model does not explicitly incorporate F-actin dynamics. Computational studies on cytoskeletal remodeling have been extensively explored in the literature, and these sources must be acknowledged and integrated into the manuscript. The authors may refer to the following studies:

• Predicting YAP/TAZ nuclear translocation in response to ECM mechanosensing.

• Directed cell migration towards softer environments.

Reviewer #2: The Authors propose a model describing the two-way feedback between FAK and mechanical characteristics of the environment. Though the subject is interesting and the attention to mechanotransduction is growing the model presents several unclear points.

In fact, in describing the evolution of the concentrations of active and inactive FAK (*ϕ*_*a*_ and *ϕ*_*d*_, respectively) and of active RhoA (*ρ*_*a*_) the model presents the following weak points

1- It is stated that "inactive species are assumed to be cytoplasm resident and activated forms membrane resident". However, while it appears that the latter actually evolves on the membrane Gamma, FAK seems to live in the cytoplasm in both forms.

2- This confusion transfers to the figures, where it is not clear what is shown. For instance, in the 2D case, is Fig.1 showing a section of the cell, or the membrane with an axisymmetric geometry? And then what about Fig.2? Ans so on ....

3- The fact that an evolution equation depends on an initial value is odd (say last equation in (1)). Since the total mass of rhoA seems to be conserved, it is better to refer to such a quantity, that is related to rho_d^0+rho_a^0/n_r.

4- A big problem regards the dimensions of the quantities introduced. This makes unclear what are the parameters actually used. For instance,

4.1- rhoA and phi are said to be concentrations, but the former is measured in #/m^2 and the latter in moles (which is not a concentration)

4.2- In Eq.(2) the dimensions of *k*_8_ will depend on the power p, as well as the dimensions of *k*_7_ in order to eventually have an *E*_*c*_ that is measured in Pascal. Certainly *k*_7_ and *k*_8_ are not *s*^-1^ as stated in Table 2. However, it would be better to define *E*_*c*_ (that I believe is then called f) as *E*_*c*_ = *k*_7_(1+(*k*_8_ ϕ_*a*_)^*p*^) where the dimensions of *k*_8_ are the inverse of those of *ϕ*_*a*_

4.3- C_1 seems to be a pure number but its dimensions are 1/(Pa s). Then being either 0 or 1 like an on-off switch makes little sense. There must be a parameter measuring the stress-induced activation of FAK.

Regarding the presentation of the results

5- The presentation of the figures can be improved. In addition to the confusion mentioned above, I suggest to replace C=0 and C=1 with a more descriptive notation, such as *σ* ↛ FAK and *σ* → FAK. Similarly, I would use FAK ↛ *E* and FAK → *E* for the columns. Then, does *E*_*c*_ = 0.6 corresponds to the parameters used for f when the effect of FAK on the Young modulus is switched off?

6- In addition, graphically speaking, while one figure presents both concentrations (why not phi_d?) the other presents the deformation only (with some confusion induced by a reordering of the "table" of cases). Would it be possble to use the same format for all variables? Or put phi, rho, and u for instance one below the other for the different cases? Or split in three figures one per state variable?

7- The so-called lamellipodium configuration is not clear starting from its introduction at line 177. In addition, the results of the simulations give an axisymmetric distribution of the concentrations and deformations that are not axisymmetric. Why is that? Are the Authors sure they are using the correct boundary conditions on the sector they study?

Finally, as a minor comment, in Eq.(5) \cdot is used to mean different operations, a scalar product between two vectors and the multiplication of a matrix (the stress) by a vector (the normal). The following sentence is a bit obscure as it is written. After all, Pi_r is just the shear stress at the surface.

For the above reasons the paper can not be accepted for publication, but because of the novelty of the topic and the approach I think it can be reconsidered for publication after a strong and careful revision.

**Have the authors made all data and (if applicable) computational code underlying the findings in their manuscript fully available?**

Reviewer #1: Yes

Reviewer #2: None

PLOS authors have the option to publish the peer review history of their article (what does this mean?). If published, this will include your full peer review and any attached files.

Reviewer #1: No

Reviewer #2: No

**Figure resubmission:**
---

## [Decision Letter · Decision Letter 1]

30 Jun 2025

PCOMPBIOL-D-24-02090R1

Mathematical modelling of mechanotransduction via RhoA signalling pathways

PLOS Computational Biology

Dear Dr. Verhees,

Thank you for submitting your manuscript to PLOS Computational Biology. After careful consideration, we feel that it has merit but does not fully meet PLOS Computational Biology's publication criteria as it currently stands. Therefore, we invite you to submit a revised version of the manuscript that addresses the points raised during the review process.

Please submit your revised manuscript within 30 days Aug 30 2025 11:59PM. If you will need more time than this to complete your revisions, please reply to this message or contact the journal office at ploscompbiol@plos.org. Please include the following items when submitting your revised manuscript:

We look forward to receiving your revised manuscript.

Kind regards,

Jochen Hub

Academic Editor

PLOS Computational Biology

Feilim Mac Gabhann

Editor-in-Chief

PLOS Computational Biology

**Additional Editor Comments (if provided):**

The reviewers acknowledged that the manuscript has been greatly improved and recommended it for acceptance.

Reviewer 2 suggests two additions that would further improve the clarity of the manuscript. Please indicate whether you will follow this recommendation or whether you disagree. Further review will not be needed, irrespective of your decision.

**Journal Requirements:**

**Reviewers' comments:**

Reviewer's Responses to Questions

**Comments to the Authors:**

Reviewer #1: no further comments

Reviewer #2: The Authors considerably improved the paper answering all question.

I only have two suggestions:

What about insert a sketch of the protein relations and their location at the beginning of the modeling section?

What about recalling that (5) represents a shear stress-free condition?

**Have the authors made all data and (if applicable) computational code underlying the findings in their manuscript fully available?**

Reviewer #1: None

Reviewer #2: None

PLOS authors have the option to publish the peer review history of their article (what does this mean?). If published, this will include your full peer review and any attached files.

Reviewer #1: No

Reviewer #2: No

**Figure resubmission:**
---

## [Editor Report · Decision Letter 2]

9 Jul 2025

Dear Ms Verhees,

We are pleased to inform you that your manuscript 'Mathematical modelling of mechanotransduction via RhoA signalling pathways' has been provisionally accepted for publication in PLOS Computational Biology.

Best regards,

Jochen Hub

Academic Editor

PLOS Computational Biology

Feilim Mac Gabhann

Editor-in-Chief

PLOS Computational Biology

---

## [Editor Report · Acceptance letter]

PCOMPBIOL-D-24-02090R2

Mathematical modelling of mechanotransduction via RhoA signalling pathways

Dear Dr Verhees,

I am pleased to inform you that your manuscript has been formally accepted for publication in PLOS Computational Biology. Your manuscript is now with our production department and you will be notified of the publication date in due course.

With kind regards,

Zsuzsanna Gémesi
